# Relationship Between the Host Plant Range of Insects and Symbiont Bacteria

**DOI:** 10.3390/microorganisms13010189

**Published:** 2025-01-17

**Authors:** Doudou Ge, Chongwen Yin, Jiayu Jing, Zhihong Li, Lijun Liu

**Affiliations:** 1College of Plant Protection, China Agricultural University, Beijing 100193, China; 15901466915@163.com (D.G.); insiluvslotus@163.com (C.Y.); jiayu@cau.edu.cn (J.J.); lizh@cau.edu.cn (Z.L.); 2Key Laboratory of Surveillance and Management for Plant Quarantine Pests, Ministry of Agriculture and Rural Affairs, Beijing 100193, China; 3Sanya Institute, China Agricultural University, Sanya 572025, China

**Keywords:** host plant range, gut symbiont bacteria, insects, tephritidae

## Abstract

The evolution of phytophagous insects has resulted in the development of feeding specializations that are unique to this group. The majority of current research on insect palatability has concentrated on aspects of ecology and biology, with relatively little attention paid to the role of insect gut symbiotic bacteria. Symbiont bacteria have a close relationship with their insect hosts and perform a range of functions. This research aimed to investigate the relationship between insect host plant range and gut symbiotic bacteria. A synthesis of the extant literature on the intestinal commensal bacteria of monophagous, oligophagous, and polyphagous tephritids revealed no evidence of a positive correlation between the plant host range and the diversity of larval intestinal microbial species. The gut symbionts of same species were observed to exhibit discrepancies between different literature sources, which were attributed to variations in multiple environmental factors. However, following beta diversity analysis, monophagy demonstrated the lowest level of variation in intestinal commensal bacteria, while polyphagous tephritids exhibited the greatest variation in intestinal commensal bacteria community variation. In light of these findings, this study proposes the hypothesis that exclusive or closely related plant hosts provide monophagy and oligophagy with a stable core colony over long evolutionary periods. The core flora is closely associated with host adaptations in monophagous and oligophagous tephritids, including nutritional and detoxification functions. This is in contrast to polyphagy, whose dominant colony varies in different environments. Our hypothesis requires further refinement of the data on the gut commensal bacteria of monophagy and oligophagy as the number of species and samples is currently limited.

## 1. Introduction

The majority of insects are herbivorous, typically consuming only fresh plant material [1,2]. The degree of specialization exhibited by herbivorous insects in their interaction with host plants can be classified into three distinct categories according to their feeding habits. Herbivores that have narrow host plants within a genus are defined as monophagous insects. In contrast, polyphagous insects exhibit a broad host range, encompassing two or more plant families. The host plants of oligophagous insects are limited to two or more genera within a family or closely related families [3]. Other researchers have employed the terms “generalist herbivores” and “specialist herbivores” to describe the feeding habits of these insects [4,5]. It is notable that the host range of herbivores is diverse, even within the same family or genus. To illustrate, the monophagous *Trirhabda canadensis* (Kirby, 1837) and the oligophagous *Cerotoma trifurcate* (Forster, 1771) and *Diabrotica virgifera* (LeConte, 1868) are members of the same family, the Chrysomelidae family (Coleoptera) [6]. The Tephritidae (Diptera) family is distributed globally and comprises approximately 5000 species [7,8]. This family provides an excellent model for the study of insect feeding habits, which encompass monophagy, oligophagy, and polyphagy.

The comparative study of monophagous, oligophagous, and polyphagous insects has been a topic of interest within the fields of ecology and biology. In terms of population size, the comparison of the numerical abundance of the insects in question helped elucidate that insects with a more general diet were represented by larger populations than those with more restricted diets under primitive conditions [9]. It was also reported that generalist and specialist herbivores preferred different host plant tissues. The generalist herbivores preferred mature leaves, while specialist herbivores favored young leaves [10]. The correlation between the nutritional composition of the host plant and the survival of tephritid species was also demonstrated. Compared to oligophagous fruit flies, polyphagous fruit flies exhibit superior survival rates in the fruits with high concentrations of carbohydrates, fiber, and lipids [11]. The feeding habits of insects are also thought to be influenced by vision. It has been proposed that monophagous or oligophagous insects may be visual specialists, whereas polyphagous insects are more likely to be visual generalists [12]. In the context of environmental change, there has been one report that investigated the impact of habitat loss on the density of butterflies with different host ranges, and they showed that the density of monophagous butterflies increased, whereas the density of other butterfly species decreased [13]. The polyphagous species of the Tephritidae family are provided with stronger invasive abilities than monophagous and oligophagous species and have had successful invasion cases [14,15]. Indeed, host plant specialization has led to significant differences in insect biology. While insect feeding habits and the evolution of symbionts have occurred simultaneously over time, these two factors are closely linked. Therefore, it is necessary to discuss the relationship between the host feeding range of insects and their symbionts.

Symbiosis is when different organisms ‘live together’. Symbiont interactions with hosts have been classified as mutualistic, parasitic, or commensal [16]. Symbionts form associations with their partners for the majority of their life history [17]. For humans, a healthy body requires the interaction of microorganisms with the host immune system. Low microbial diversity has been linked to a variety of diseases [18]. Microorganisms inhabit plant roots and become root symbionts, which are critical to host adaptation and impact plant productivity and health [19,20,21]. For insects, the symbiosis between insect host and microbe has had a profound impact on the evolution of life [22,23]. Symbionts not only impact their host fitness but also shape the trajectory of their phenotype [24]. Therefore, it is important to explore the relationship between microorganisms and their hosts in studies related to insects.

Bacteria are a diverse, abundant, and ubiquitous group of organisms, exhibiting a range of characteristics and functions. Insects are host to both obligate and facultative symbionts. The former provide essential nutrients required by the host and reside in specialized cells (bacteriocytes) [25,26,27]. Facultative symbionts are not essential for insect survival and reproduction [28], yet they perform other functions [29,30]. Facultative symbionts have been observed to confer resistance to entomopathogenic fungi and parasitoid wasps, enhance the detrimental effects of heat, and influence host plant adaptability to the insect host [28,31,32]. Furthermore, these microorganisms can be distinguished as either intracellular or extracellular based on their location within insect cells [33,34]. In some cases, the locations of bacteria are flexible and mobile. For example, “*Candidatus* Erwinia dacicola” has been found in all life stages of *Bactrocera oleae* (Rossi, 1790). It is one of the few non-pathogenic endosymbionts that is capable of undergoing a conversion between intracellular and extracellular lifestyles [35,36].

The microbiota associated with phytophagous insects performs a variety of functions that facilitate the exploitation of plant resources [37]. The roles of symbiont bacteria have been a subject of extensive research from a multitude of perspectives. The gut bacteria of insects have been demonstrated to contribute to a number of processes, including nutrition, immune response modulation, protection from parasites and pathogens, and communication [22,25,38,39]. It is therefore necessary to investigate the role of insect symbiotic bacteria in insect feeding habits.

The main objective of this study was to investigate the relationship between insect host plant range and gut symbiotic bacteria. To achieve this, the information on gut symbiotic bacteria in tephritids was summarized. We compared the alpha diversity of symbiont bacteria of tephritids with different plant specializations and explored the causes of variation in gut commensal bacterial communities and the presence of core bacterial groups.

## 2. Relationship Between the Host Plant Range of Tephritids and Symbiont Bacteria

### 2.1. Is the Narrowing of the Insect’s Dietary Range Accompanied by a Reduction in the Diversity and Variation in Gut Bacteria?

In the order Diptera, it has been demonstrated that microbial communities in polyphagous species are more diverse than those in monophagous species. The gut bacteria diversity of monophagous *Bactrocera cacuminata* (Hering, 1941) was found to be inferior to that of the polyphagous species, including *Bactrocera tryoni* (Froggatt, 1897), *Bactrocera neohumeralis* (Hardy, 1951), and *Bactrocera jarvisi* (Tryon, 1927) [40]. In the order Hemiptera, the low diversity of symbionts associated with the oligophagous aphid *Aphis citricidus* (Kirkaldy, 1907) was elucidated due to the absence of two common aphid-associated secondary symbionts, namely *Hamiltonella defensa* and *Regiella insecticola* [41,42]. Nevertheless, no discernible correlations between fruit fly dietary breadth and microbiome diversity or abundance patterns have been identified [43].

In order to ascertain the veracity of this hypothesis, this study tested it with Tephritidae insects comprising monophagous, oligophagous, and monophagous species. A total of 19 articles were identified as relevant for the purposes of this study and were included in this review. These articles presented research on the composition of the gut symbiotic bacteria in larvae and included a total of 20 species. Of the species examined, *B. oleae* (BO) was found to be monophagous. The oligophagous species included *Bactrocera minax* (Enderlein, 1920) (BM), *Ceratitis podocarpi* (Bezzi, 1924) (CP), and *Zeugodacus cucurbitae* (Coquillett, 1899) (ZC). Polyphagous species, including *Anastrepha ludens* (Loew, 1873) (AL), *Anastrepha obliqua* (Macquart, 1835) (AO), *Anastrepha serpentina* (Wiedemann, 1830) (ASE), *Anastrepha striata* (Schiner, 1868) (AST), *Anastrepha fraterculus* (Wiedemann, 1830) (AF), *Anastrepha grandis* (Macquart, 1846) (AG), *Bactrocera dorsalis* (Hendel, 1912) (BD), *Bactrocera zonata* (Saunders, 1842) (BZ), *Bactrocera tryoni* (Froggatt, 1897) (BT), *Zeugodacus tau* (Walker, 1949) (ZT), *Bactrocera carambolae* (Drew and Hancock, 1994) (BC), *Ceratitis capitate* (Wiedemann, 1824) (CC), *Ceratitis quilicii* (De Meyer, Mwatawala and Virgilio, 2016) (CQ), *Ceratitis rosa* (Karsch, 1887) (CR), *Ceratitis cosyra* (Walker, 1849) (CCO), and *Ceratitis flexuosa* (Walker, 1853) (CF), were included in this study (Table 1). The sequencing methods employed for the analysis of gut commensal bacteria include pyrophosphate sequencing and Illumina sequencing. The samples included both field-collected and laboratory samples. The host plants and the collection sites of the tephritid species exhibit considerable variation (Table 2). A total of 40 sequencing results for 20 species were re-pumped flat using the vegan package (2.6-4) in R (4.3.1). The resulting alpha diversity indices are presented in Figure 1.

**Table 1 microorganisms-13-00189-t001:** Sequencing data on the intestinal commensal bacteria of tephritid larvae from different studies.

Name	Species	Sequencing Method	Source
1AL	*Anastrepha ludens* (Loew, 1873)	Pyrosequencing	[44]
1AO	*Anastrepha obliqua* (Macquart, 1835)	Pyrosequencing	[44]
1ASE	*Anastrepha serpentina* (Wiedemann, 1830)	Pyrosequencing	[44]
1AST	*Anastrepha striata* (Schiner, 1868)	Pyrosequencing	[44]
2AF	*Anastrepha fraterculus* (Wiedemann, 1830)	Illumina	[45]
3AO	*Anastrepha obliqua* (Macquart, 1835)	Metatranscriptomic	[37]
4AG	*Anastrepha grandis* (Macquart, 1846)	Pyrosequencing	[46]
4AL	*Anastrepha ludens* (Loew, 1873)	Pyrosequencing	[46]
4BO	*Bactrocera oleae* (Rossi, 1790)	Pyrosequencing	[46]
5CC	*Ceratitis capitata* (Wiedemann, 1824)	Illumina	[47]
6CCW	*Ceratitis capitata* (Wiedemann, 1824)	Illumina	[48]
6CCL	*Ceratitis capitata* (Wiedemann, 1824)	Illumina	[48]
7BM	*Bactrocera minax* (Enderlein, 1920)	Pyrosequencing	[49]
8BM	*Bactrocera minax* (Enderlein, 1920)	Metagenomic	[50]
9BO	*Bactrocera oleae* (Rossi, 1790)	Illumina	[51]
10ZC	*Zeugodacus cucurbitae* (Coquillett, 1899)	Illumina	[52]
11BD1	*Bactrocera dorsalis* (Hendel, 1912)	Illumina	[53]
11BD2	*Bactrocera dorsalis* (Hendel, 1912)	Illumina	[53]
11BD3	*Bactrocera dorsalis* (Hendel, 1912)	Illumina	[53]
12BD	*Bactrocera dorsalis* (Hendel, 1912)	Pyrosequencing	[54]
13BD	*Bactrocera dorsalis* (Hendel, 1912)	Illumina	[55]
14BD	*Bactrocera dorsalis* (Hendel, 1912)	Illumina	[43]
14ZC	*Zeugodacus cucurbitae* (Coquillett, 1899)	Illumina	[43]
14BO	*Bactrocera oleae* (Rossi, 1790)	Illumina	[43]
14BZ	*Bactrocera zonata* (Saunders, 1842)	Illumina	[43]
14CC	*Ceratitis capitata* (Wiedemann, 1824)	Illumina	[43]
14CQ	*Ceratitis quilicii* (De Meyer, Mwatawala and Virgilio, 2016)	Illumina	[43]
14CR	*Ceratitis rosa* (Karsch, 1887)	Illumina	[43]
14CCO	*Ceratitis cosyra* (Walker, 1849)	Illumina	[43]
14CF	*Ceratitis flexuosa* (Walker, 1853)	Illumina	[43]
14CP	*Ceratitis podocarpi* (Bezzi, 1924)	Illumina	[43]
15BT	*Bactrocera tryoni* (Saunders, 1842)	Illumina	[56]
16BT1	*Bactrocera tryoni* (Saunders, 1842)	Illumina	[56]
16BT2	*Bactrocera tryoni* (Saunders, 1842)	Illumina	[56]
16BT3	*Bactrocera tryoni* (Saunders, 1842)	Illumina	[56]
16BT4	*Bactrocera tryoni* (Saunders, 1842)	Illumina	[56]
16BT5	*Bactrocera tryoni* (Saunders, 1842)	Illumina	[56]
17ZT	*Bactrocera tau* (Saunders, 1842)	Illumina	[57]
18BZ	*Bactrocera zonata* (Saunders, 1842)	Illumina	[58]
19BC	*Bactrocera carambolae* (Drew and Hancock, 1994)	Illumina	[59]

The alpha diversity of intestinal commensal bacteria did not demonstrate an upward trend in conjunction with an increase in the feeding range (Figure 1). Instead, the data demonstrated a high degree of volatility. Alpha diversity index of the same species demonstrated considerable fluctuations in the sequencing results across the various articles. To illustrate, the diversity of 9BO was markedly greater than that of 4BO and 14BO in samples of intestinal commensal bacteria from *B. oleae*. A similar result was observed in *B. minax*, *A. ludens*, *A. obliqua*, *B. zonata*, *C. capitata*, *Z. tau*, and *B. dorsalis*.

Following the calculation of Bray–Curtis distances, NMDS and PCoA analyses were conducted to elucidate the discrepancies between the commensal bacterial communities (Figure 2). The data from the wild samples were selected for inclusion in this study. The research objects were divided into six groups, comprising *B. oleae* (BO), *B. minax* (BM), *Z. cucurbitae* (ZC), *B. dorsalis* (BD), *C. capitata* (CC), and *B. tryoni* (BT). Each data point within the same group was derived from a different source of literature. This indicates that alterations to the host plant, geographical environment, and sequencing methods may result in discrepancies in the intestinal symbiotic community of the same tephritid. The NMDS analysis yielded a stress value of less than 0.2, which lends credence to the reliability of the results. The results demonstrated that the distance between samples within the monophagous group BO was the shortest, followed by the samples within the oligophagous groups BM and ZC. The distance between samples within the polyphagous BD, CC, and BT groups was found to be the longest. The PCoA analysis revealed that 38.2% of the observed variation can be attributed to the differences between the samples. The distance between samples within the monophagous group was also found to be smaller than that observed between samples in the oligophagous and polyphagous groups. This suggests that the intestinal symbiotic community of monophagous tephritids is less susceptible to variation in response to different environmental factors. In contrast, the intestinal symbiotic flora of polyphagous tephritids exhibited significant alterations.

### 2.2. Factors Contributing to Differences in Intestinal Commensal Bacteria in the Same Species of Tephritidae

The latest research indicates that a number of factors influence the intestinal symbionts of insects [60]. A variety of factors, including insect species, geography, host plants, developmental stages, and sequencing methods, can give rise to notable variations in insect gut symbiotic bacterial communities. The interaction of biotic and abiotic factors results in alterations to the gut commensal bacteria. In terms of abiotic factors, temperature can influence the abundance of bacteria within the host or their efficiency of transmission to the offspring [61]. Nevertheless, the impact of plant hosts and geography on gut symbiotic bacteria has been the subject of the most extensive research in the context of insect larvae. The symbiotic bacterial communities are shaped by both the host and the habitat, but the specific ways in which this occurs vary [62].

Geography represents a significant factor influencing the structure of symbiont communities. A notable discrepancy was observed between peach samples from Concordia and Horco Molle in the case of *A. fraterculus*. The fauna was distinct and represented a biological factor that gives rise to alterations in the gut microbiology [45]. In *B. dorsalis*, the composition of the bacterial community varied according to the geographical location of the populations. The number of bacterial families in the gut microbiome of the Hainan population was lower than that of the Guizhou population [63]. In *Z. cucurbitae*, bacterial diversity and abundance varied within and between samples from three geographical regions, namely Thailand, Peninsular Malaysia, and Sarawak [64]. It has been demonstrated that geography can alter the structure of symbiont communities in two oligophagous aphids, namely *Melanaphis sacchari* (Zehntner, 1897) and *Neophyllaphis podocarpi* (Takahashi, 1920) [65]. Meanwhile, facultative symbionts displayed distinctive geographic distribution patterns in *Acyrthosiphon pisum* (Harris, 1776) [66]. The research project concentrated on the symbionts of *Aphis gossypii* (Glover, 1877) and revealed significant differences between the Japanese and Australian samples [67]. There was a comprehensive overview working on the geographic distribution of aphid secondary symbionts, with a particular focus on the dominant symbionts across different continents [68]. Notable variations were found in the bacterial communities of two parthenogenetic populations collected from distinct islands in Hawaii. This underscores the significant influence of geographical distribution on the symbiotic microorganisms of aphids [69]. Additionally, the symbiotic bacteria of a monophagous caterpillar, *Tyria jacobaeae* (Linnaeus, 1758), also exhibited similarities to soil symbiotic bacteria in its habitat [70].

The host plant is another factor that can exert an influence on insect symbionts. The bacterial profiles of *A. fraterculus* larvae collected from guavas and peaches were found to be different. The gut microbiota of larvae collected from guavas exhibited greater richness, phylogenetic diversity, equitability, and lower dominance than that of larvae collected from peaches [45]. The bacterial communities in *A. obliqua* appear to be structured according to the insect life stage and the host plant [71]. Samples of *C. capitata* larvae were collected from figs, cacti, peaches, fenugreek, and citrus. Sequencing results indicate that changes in the host plant can be the cause of changes in the gut symbiotic bacteria [47]. Changes in the gut microbiota influenced by host fruits were similarly found in *B. dorsalis* [72]. In aphids, the host plant is identified as a significant factor in structuring bacterial communities, exerting selection pressure [73,74,75]. In *Aphid craccivora* (Koch, 1854), the food plant of the aphid had a significant association with the secondary symbiont, *Arsenophonus*, which was found to have a high prevalence in aphids collected from locusts. In contrast, *Hamiltonella* was dominant in the alfalfa populations [76]. The transition from a hobby host plant to a general host plant resulted in a notable decline in fitness for oligophagous aphids, in comparison to the polyphagous aphid. The density of *Buchnera* increased with the development of the *Aphid citricidus* (Kirkaldy, 1907), an oligophagous aphid, on sweet oranges, but decreased with the development of the same aphid on orange jasmines. In contrast, the polyphagous *Aphid aurantii* (Fonscolombe, 1841) showed an increasing trend in *Buchnera* density in both host plants [77]. The results demonstrated that the structure and activity of the microbial communities exhibited notable differences between infested plants [37,71]. One reason for the change in bacteria community is the presence of certain marker microorganisms in the host plant [78]. Another is that the larvae select for specific bacteria that play a role in supplementing the nutrient deficiencies of the plant host for the insect or degrading the toxic compounds of the plant host, while they feed on special plant material [79]. It has been proposed that the composition of microbial communities may be influenced by the diets of insects [40].

### 2.3. Core Flora Was Present in Monophagous and Oligophagous Insects

The heterogeneity of the field samples collected for a given study makes it challenging to exercise complete control over the variables. For instance, investigations into the impact of diverse geographical locations on intestinal symbiotic bacteria frequently coincided with variations in the plant hosts under consideration. This phenomenon is particularly evident in polyphagous insects. However, the gut symbiotic bacteria of monophagous insects are relatively stable in the host plant, thus allowing for more effective control of variables. This may also be the reason why the differences in gut commensal bacteria between monophagous and oligophagous tephritids are smaller than those of polyphagous tephritids in different environments. The long-term symbiosis between insects and intestinal symbiotic bacteria on the same host plant results in the establishment of a stable relationship.

A significant number of studies have indicated the existence of a robust symbiotic relationship between the olive fruit fly in the field and *Candidatus* Erwinia dacicola [80,81,82]. The symbiont is transferred vertically to the offspring through the contamination of the egg surface, the deposition of bacterial capsules on eggs, the consumption of the mother’s excrement, or through trans-ovarial transmission [83]. There is a paucity of research on other monophagous insects in Tephritidae, but there is a high probability of a stable symbiotic relationship. In the case of oligophagous insects, the measured intestinal commensal bacteria of *B. minax* exhibited a high abundance of *Klebsiella* [49,50]. The supplementation of adult diets with *Klebsiella* bacterial isolates resulted in a significant enhancement of female fecundity [84]. Additionally, *Citrobacter* was postulated to be the dominant bacterial species in the adult *B. minax* population [85]. However, the research indicated that the abundance of this bacteria is low in larvae [49]. *Klebsiella*, *Acinetobacter*, and *Providencia* were identified in known samples of the intestinal tract of *Z. cucurbitae* larvae, but their abundance was not found to be consistent [43,52]. In the case of the typical polyphagous *B. dorsalis* and *C. capitata*, no particular symbiotic bacterium was identified as being common to all samples, and the symbiotic communities were found to be highly variable across different environments.

The core flora found in monophagous and oligophagous tephritids may fulfill nutritional and detoxification functions which are closely related to the plant hosts. The majority of aphids in the Hemiptera order form deep associations with *Buchnera*, including those that are oligophagous. *Buchnera* provides essential amino acids to aphids, which are exchanged for non-essential amino acids [34,86,87,88]. The aphid host provides essential amino acids to this endosymbiont, whose genome has undergone erosion [77,89,90,91]. In the planthopper *Ommatidiotus dissimilis* (Fallén, 1806), the role of *Sodalis*-like bacteria has been postulated to be involved in the provision of essential nutrients to the host insect [92]. In the brown planthopper, microbial symbionts provide the genes necessary for insects to survive in an unbalanced diet and to conduct conservative biochemical pathways [93].

Intestinal symbionts have been demonstrated to play a role in detoxification in some monophagous insects. In *B. oleae*, the symbiont *Candidatus* Erwinia dacicola has been observed to offset the inhibitory effects of oleuropein, a principal phenolic glycoside present in unripe olives. This finding has led to the conclusion that this symbiont is essential for the development of *B. oleae* [80,81,82,94]. The olive moth, *Prays oleae* (Bernard, 1788), is a monophagous insect that feeds on olive trees. It has been observed that intestinal bacteria, specifically *Acetinobacter* sp. or *Staphylococcus* sp., play a role in detoxification, enabling the moth to overcome the secondary metabolites produced by olive trees [95]. The study of this symbiotic relationship can therefore provide a basis for the control of *B. oleae* [96,97]. In the order Hemiptera, the insecticide-resistant nymphs of the rice stink bug, *Nilaparvata lugens* (Stål, 1854), were found to be enriched with bacteria belonging to the Burkholderiales which possess detoxification functions [98]. Due to the disparate host ranges of insects, the symbiont has undergone significant specialization to overcome the detrimental effects of a fixed substance.

The current study has relatively limited data on larval gut microbial sequencing for *B. oleae*, *B. minax*, and *Z. cucurbitae*, and there is a paucity of results on gut microbial sequencing for the polyphagous fruit fly. In particular, only one monophagy, *B. oleae*, has been shown to harbor intestinal symbiotic bacteria in Tephritidae. Furthermore, the number of sample replicates for monophagous and oligophagous tephritids is relatively limited, which has an impact on the veracity of our hypothesis. It is therefore necessary to obtain further sequencing results of the intestinal commensal bacteria of monophagous and oligophagous tephritids in order to verify whether monophagy will establish stable relationships with specific intestinal commensal bacteria, which is difficult to do in polyphagy.

## 3. Conclusions

In conclusion, this paper focused on insects of the Diptera family Tephritidae and summarized the sequencing results of gut symbiotic bacteria from the current articles. There was no significant correlation between the diversity of gut commensal bacteria and the dietary range of tephritids, but the analysis of community differences revealed that monophagy showed the least variation in gut commensal bacteria across environments, followed by oligophagy, and the greatest variation in gut commensal bacteria was found in polyphagy. In this regard we suggested that this phenomenon may be due to the existence of core gut flora in monophagous and oligophagous tephritids that arose from their long-term evolution with plant hosts. This core flora has a role in providing nutrients and detoxification and is closely related to the plant hosts. However, since there is still a paucity of published sequencing data on oligophagous and monophagous insect gut symbionts, subsequent validation is needed.

## Figures and Tables

**Figure 1 microorganisms-13-00189-f001:**
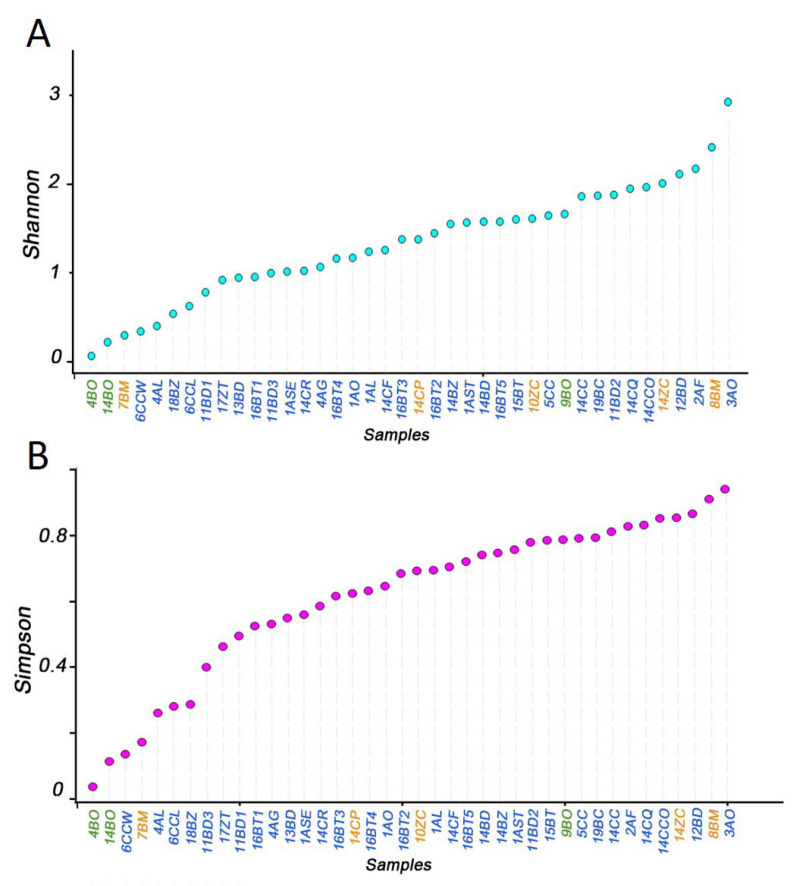
Intestinal bacteria diversity for 20 species of tephritidae. (**A**) Shannon index. (**B**) Simpson index. Monophagy (green), oligophagy (orange), and polyphagy (blue). BO: *Bactrocera oleae*; BM: *Bactrocera minax*; CP: *Ceratitis podocarpi*; ZC: *Zeugodacus cucurbitae*; ASE: *Anastrepha serpentine*; AST: *Anastrepha striata*; AL: *Anastrepha ludens*; AO: *Anastrepha obliqua*; AF: *Anastrepha fraterculus*; AG: *Anastrepha grandis*; ZT: *Zeugodacus tau*; BZ: *Bactrocera zonata*; BC: *Bactrocera carambolae*; BT: *Bactrocera tryoni*; CC, CCW, and CCL: *Ceratitis capitata*; CCO: *Ceratitis cosyra*; CQ: *Ceratitis quilicii*; CR: *Ceratitis rosa*; CF: *Ceratitis flexuosa*; BD: *Bactrocera dorsalis*.

**Figure 2 microorganisms-13-00189-f002:**
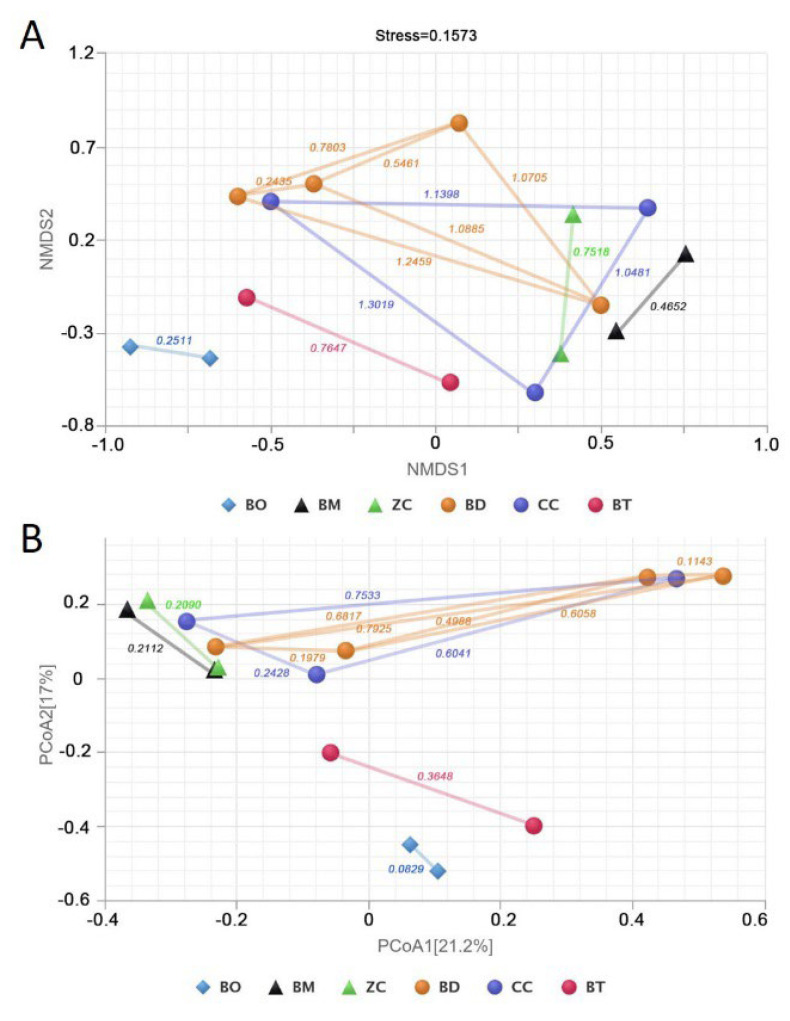
β diversity analysis of bacteria community. (**A**) NMDS analysis. (**B**) PCoA analysis. BO: Bactrocera oleae; BM: Bactrocera minax; ZC: Zeugodacus cucurbitae; BD: *Bactrocera dorsalis*; CC: *Ceratitis capitata*; BT: *Bactrocera tryoni*.

**Table 2 microorganisms-13-00189-t002:** Information on samples of intestinal commensal bacteria of tephritid larvae from different studies.

Name	Wild/Laboratory	Host Plant	Geography
1AL	Wild	Bitter orange (*Citrus aurantium*)	Soconusco Region, Chiapas State, Mexico
1AO	Wild	Mango (*Mangifera indica*),	Soconusco Region, Chiapas State, Mexico
1ASE	Wild	Mamey sapote (*Pouteria sapota*)	Soconusco Region, Chiapas State, Mexico
1AST	Wild	Guava (*Psidium guajava*)	Soconusco Region, Chiapas State, Mexico
2AF	Wild	Peaches and guavas	Horco Molle, Tucuman Province, Argentina; Concordia, Entre Rios Province, Argentina
3AO	Wild	*Spondias purpurea*, *Mangifera indica*, and *Averrhoa carambola*	Valle del Cauca, southwestern Colombia
4AG	Laboratory	/	/
4AL	Laboratory	/	/
4BO	Laboratory	/	/
5CC	Wild	Orange fruits (*Citrus sinensis*)	Reggio Calabria, Italy
6CCW	Wild	Mandarin orange (*Citrus reticulada* and *Citrus unshiu*)	Valencia, Spain
6CCL	Laboratory	/	/
7BM	Wild	Citrus	Yichang, Hubei, China
8BM	Wild	Citrus	Yichang, Hubei, China
9BO	Wild	Olives	Greece
10ZC	Wild	Cucumber (*Cucumis sativus*)	Farming Systems Research Center for Hill and Plateau Region, Ranchi, India
11BD1	Laboratory	/	/
11BD2	Wild	Carambola (*Averrhoa carambola*)	Huizhou, Guangdong, China
11BD3	Wild	Carambola (*Averrhoa carambola*)	Nansha, Guangdong, China
12BD	Wild	Unknown	Wuhan, Hubei, China
13BD	Wild	Carambola (*Averrhoa carambola*)	Guangzhou, Guangdong, China
14BD	Wild	*Eriobotrya japonica*, *Mangifera indica*, *Annona muricata*, and *Psidium guajava*	South Africa; Tanzania
14ZC	Wild	*Coccinia grandis*, *Momordica charantia*, *Citrullus lanatus*, and *Cucumis sativus*	Reunion; Tanzania
14BO	Wild	*Olea europea*	Greece; Italy
14BZ	Wild	*Terminalia catappa*	Reunion
14CC	Wild	*Citrus reticulata*, *Ficus carica*; *Malus pumila*, *Ficus carica*; *Pyrus communis*	Greece; Italy; South Africa
14CQ	Wild	*Eriobotrya japonica*, *Psidium catlleyanum*, *Psidium guajava*, *Eriobotrya japonica*, and *Harpephyllum caffrum*	Reunion; South Africa
14CR	Wild	*Citrus sinensis*	Mozambique
14CCO	Wild	*Sclerocarya birrea* and *Annona muricata*	Tanzania; South Africa
14CF	Wild	*Antiaris toxicaria*	Kenya
14CP	Wild	*Afrocarpus falcatus*	South Africa
15BT	Wild	Pomegranates (*Punica granatum*), green apples (*Malus pumila*), and quinces (*Cydonia oblonga*)	New South Wales and Victoria, Australia
16BT1	Wild	Hog plum	Nambour
16BT2	Wild	Sapodilla	Whiteside
16BT3	Wild	Sapodilla	Nambour
16BT4	Wild	Pomegranate	Commealla
16BT5	Wild	Green apple	Echuca
17ZT	Laboratory	/	/
18BZ	Wild	Wood apple (*Aegle marmelos*)	Research farm of ICAR Research Complexfor Eastern Region, Ranchi, India
19BC	Wild	Carambola (*Averrhoa carambola*)	Wilayah Persekutuan Kuala Lumpur, Malaysia

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
