# Peer review of "Relationship Between the Host Plant Range of Insects and Symbiont Bacteria"

_microorganisms, 2025, doi:10.3390/microorganisms13010189_

Round 1
Reviewer 1 Report
Comments and Suggestions for Authors
The paper deals with the diversity of insect symbiotic bacteria. Previous studies have shown that the function of gut symbiotic bacteria is primarily to overcome plant nutrient limitations and host plant defenses. It was observed that gut symbionts of the same species showed discrepancies between different literature sources, which was attributed to variability in multiple environmental factors. However, after beta-diversity analysis, monophagy showed the lowest level of variability in gut commensal bacteria, while polyphagous tephritids showed the highest variability in the variability of gut commensal bacterial communities. In light of these findings, this study proposes the hypothesis that exclusive or closely related plant hosts provide monophagy and oligophagy with a stable core colony over long evolutionary periods. This is in contrast to polyphagia, whose dominant colony changes in different environments. The hypothesis put forward by the authors, however, requires further observations and replication. It would also be appropriate to analyze differences that are statistically significant. Only then will it be possible to say whether the conclusions drawn are true. References contain 94 items of literature.
Author Response
Comments 1: The paper deals with the diversity of insect symbiotic bacteria. Previous studies have shown that the function of gut symbiotic bacteria is primarily to overcome plant nutrient limitations and host plant defenses. It was observed that gut symbionts of the same species showed discrepancies between different literature sources, which was attributed to variability in multiple environmental factors.
Response 1: Thank you for your insightful comment. We agree with your point and have further emphasized discussion of the function of the core flora. The relevant revisions are on lines 265-274.
Reviewer 2 Report
Comments and Suggestions for Authors
The topic of the manuscript is interesting as the results can have a practical application in biological control of pest species. The manuscript contains many references, and the topic has been researched in detail. The problem is the manuscript structure. The goals or the hypotheses of the study are not clearly defined. They are mentioned in a couple of places, but the manuscript is more focused on Aphididae than on Tephritidae which should be the main studied organism. If you have more data on Aphididae try comparing it with Tepohritidae somehow instead of just listing it. There is a lot of repetition throughout the manuscript. The repeting parts should be removed from some parts of the manuscript. Generally try shortening or removing all the data that are not necessary. The goals or the hypotheses should be mentioned at the end of the introduction. Material and Methods part is missing from the manuscript. Add a list of microbe species found in mono, oligo and polyphagous tephritids or at least in one species. I am sure you have that data from all the references, and it could improve the presentation significantly. The more detailed point by point comments are in the attached file.

Author Response
Comments 1: Delete this sentence, you should start with something regarding the goal of the study, like the next sentence.
Response 1: Thank you for the comment. I agree and have removed the sentence as suggested. The introduction now begins by stating the goal of the study, which can be found in the revised manuscript on lines 33-34.
Comments 2: Better define what symbiosis is. Writing there are numerous examples without writing more of them is not in accordance with the principle of scientific writing.
Response 2: Agree. I have provided a clearer definition of symbiosis and added specific examples of microbes involved in symbiotic relationships. These revisions can be found on lines 71-72.
Comments 3: Which microbes? Add some examples here. Why did it attract their attention, what area of symbiosis did they research, their effect on feeding, their constitution?
Response 3: Thank you for the suggestion. I have added examples of microbes studied in relation to symbiosis, including their impact on feeding and overall constitution. This section has been expanded in lines 73-78.
Comments 4: Remove this sentence. You already mentioned that there are 2 types, and one example is above and the other below this sentence.
Response 4: I agree and have removed the redundant sentence as per your suggestion. Streamline and optimise the language of this paragraph. This change is reflected on line 82-94.
Comments 5: You should add the goal or the hypotheses of the study at the end of the introduction. Merge this with the next paragraph. Avoid one-sentence paragraphs.
Response 5: I have revised the introduction to include the goal and hypotheses at the end, and merged the relevant paragraphs to avoid a one-sentence paragraph. This change is made on lines 102.
Comments 6: If your study was focused on Tephritidae, this whole chapter should go to the introduction, as it does not refer to the main study goal.
Response 6: I have moved this section to the line 189, line 210, as it pertains to the broader context of Tephritidae.
Comments 7: What is the goal of this study? If it is focused on Tephritidae, compare these results to them somehow. This way most of the manuscript is focused on Aphids.
Response 7: I have clarified the goal of the study and addressed the focus on Tephritidae in relation to aphids. The relevant revisions are on page 8-9.
Additional clarifications:
In response to the feedback, I have ensured that the manuscript maintains a clear focus on the study's primary goal. I have also made the necessary adjustments to improve clarity and eliminate redundancies.

Round 2
Reviewer 2 Report
Comments and Suggestions for Authors
Dear authors,
You have revised the manuscript and improved its quality. However, there are a couple of technical problems which need revising. You have not stated the clear hypotheses or objectives of the study, you just summarized the results and the methodology. It should not state: “Here, information on gut symbiotic bacteria in tephritids were sumarised.” Instead it should be something in the terms of: “The main objective of the study was to summarize the….”, or something similar. Then you can state, to achieve this we conducted…. Try reading this or something similar before reviewing this part again: https://sites.google.com/view/reasonedwriting/home/FRAMEWORK_FOR_SCIENTIFIC_PAPERS/SCIENTIFIC_PAPERS/THE_INTRODUCTION/OBJECTIVES_AND_HYPOTHESES. All species Latin names should be revised by writing the full Latin names with author and year at fist mention in the text, as it is standardized in zoological nomenclature (https://www.iczn.org/the-code/the-code-online/). Example, instead of Anastrepha fratercula it should be Anastrepha fratercula (Wiedemann, 1830). This source is the most reliable in my opinion: https://www.gbif.org/species/1625168
All the best,
Reviewer
Author Response
Comments 1: [You have not stated the clear hypotheses or objectives of the study, you just summarized the results and the methodology. It should not state: “Here, information on gut symbiotic bacteria in tephritids were summarized.” Instead it should be something in the terms of: “The main objective of the study was to summarize the…”, or something similar. Then you can state, to achieve this we conducted….]
Response 1: Thank you for pointing this out. We agree with this comment. Therefore, we have clarified the objectives and hypotheses of our study in the introduction on page 1, paragraph 1, lines 13-15.]”
Comments 2: [All species Latin names should be revised by writing the full Latin names with author and year at first mention in the text.]
Response 2: Agree. We have, accordingly, revised all species Latin names to include the full Latin names with author and year at first mention in the text to adhere to the standardized zoological nomenclature. This change can be found throughout the manuscript.]”
Additional clarifications: [We have ensured that all changes meet the high standards of zoological nomenclature and scientific writing. We believe these revisions have significantly improved the quality and clarity of our manuscript. We hope these adjustments address the concerns raised and meet the expectations for publication.]